# Contemporary Lifestyle and Neutrophil Extracellular Traps: An Emerging Link in Atherosclerosis Disease

**DOI:** 10.3390/cells10081985

**Published:** 2021-08-04

**Authors:** Laura Pérez-Olivares, Oliver Soehnlein

**Affiliations:** 1Center for Molecular Biology of Inflammation (ZMBE), Institute for Experimental Pathology (ExPat), Westfälische Wilhelms-Universität (WWU), 48149 Münster, Germany; soehnlein@uni-muenster.de; 2Department of Physiology and Pharmacology (FyFa), Karolinska Institute, 17165 Stockholm, Sweden

**Keywords:** neutrophils, neutrophil extracellular traps (NETs), atherosclerosis, cardiovascular risk factors, inflammation, innate immunity

## Abstract

Neutrophil extracellular traps (NETs) are networks of extracellular genetic material decorated with proteins of nuclear, granular and cytosolic origin that activated neutrophils expel under pathogenic inflammatory conditions. NETs are part of the host’s innate immune defense system against invading pathogens. Interestingly, these extracellular structures can also be released in response to sterile inflammatory stimuli (e.g., shear stress, lipidic molecules, pro-thrombotic factors, aggregated platelets, or pro-inflammatory cytokines), as in atherosclerosis disease. Indeed, NETs have been identified in the intimal surface of diseased arteries under cardiovascular disease conditions, where they sustain inflammation via NET-mediated cell-adhesion mechanisms and promote cellular dysfunction and tissue damage via NET-associated cytotoxicity. This review will focus on (1) the active role of neutrophils and NETs as underestimated players of the inflammatory process during atherogenesis and lesion progression; (2) how these extracellular structures communicate with the main cell types present in the atherosclerotic lesion in the arterial wall; and (3) how these neutrophil effector functions interplay with lifestyle-derived risk factors such as an unbalanced diet, physical inactivity, smoking or lack of sleep quality, which represent major elements in the development of cardiovascular disease.

## 1. Introduction

The contemporary lifestyle of Western societies is characterized by several behavioral and environmental cardiovascular risk factors. Hyperlipidemia, hypercholesterolemia, hyperglycemia and hypertension (when derived from an unhealthy diet), lack of physical exercise, smoking and altered or deficient sleeping patterns are all modifiable risk factors for coronary artery disease (CAD) [1]. The long-term exposure to these behavioral risk factors determines an enhanced prevalence for chronic multifactorial diseases that manifest chronic inflammation and immune dysregulation as common hallmarks. Obesity, diabetes and hypertension are conditions that ultimately lead to an increased incidence of cardiovascular disease (CVD), degenerative disorders of the central nervous system, respiratory illnesses and cancer [2,3,4]. These non-communicable diseases account for more than seventy percent of the total global deaths annually [1].

CVDs are a group of heart and blood vessels disorders that account for an estimated 18 million deaths per year globally [5]. The most life-threatening complications of CVD are myocardial infarction and stroke, which are conditions caused by the thrombotic occlusion of a coronary artery or a cerebral microvessel. The primary underlying mechanism of most cardiovascular complications is atherosclerosis, a chronic, lipid-driven and multi-factorial inflammatory disease of the arterial vessel wall that extends over time through excessive arterial leukocyte recruitment [5,6]. Among CVDs, atherosclerosis disease is the leading cause of global morbidity and mortality [7]. Briefly, atherosclerosis is characterized by a progressive endothelial dysfunction and vessel wall inflammation, subendothelial cholesterol accumulation and neointimal thickening, which results in plaque formation and consequent lumen narrowing of affected arteries [8]. Uncontrolled myeloid cell recruitment to atherosclerotic lesions is a critical step in the progression of lesion development. Monocytes, monocyte-derived macrophages and lesional macrophages further differentiate into foam cells, and comprise the predominant subset of innate immune cells in atherosclerotic lesions [9]. However, neutrophils and particularly neutrophil extracellular traps (NETs) were recently postulated as essential participants in atherogenesis and arterial wall inflammation. Namely, NETs can pave the way for monocyte recruitment or exert cytotoxicity towards smooth muscle cells (SMCs) and endothelial cells [10,11,12].

Here, we summarize the current knowledge of the inflammatory processes in atherosclerotic disease, with a particular focus on the active role of neutrophils and NETs as underestimated players on atherogenesis and lesion progression. We also critically review the available literature that brings together the causality between the long-established atherosclerotic risk factors and the NETs. Targeting inflammation and developing public health prevention-based strategies towards modifiable cardiovascular risk factors in the atherosclerotic context would substantially improve disease burden, worldwide disability rates, quality of life and ultimately, human health.

## 2. Mechanisms and Initiators of NET Formation

Neutrophils or polymorphonuclear (PMN) leukocytes are the most abundant granulocytes in human blood where they comprise around 50–70% of total circulating leukocytes. In mice, neutrophils only represent about 10% of total circulating leukocytes [13]. Neutrophils play an essential part in innate immunity as the first line of defense against invading pathogens [14], but also as key mediators of sterile tissue injury [15,16]. The homeostasis of neutrophils is maintained by balancing their short lifespan in circulation with their regulated generation from myeloid precursors and release from the bone marrow [17]. Neutrophil daily production is regulated by the interleukin (IL)-23/IL-17/G-CSF axis and can reach up to 10^^11^ cells [18,19]. Neutrophils are designated as short-lived cells with a reported half-life in circulation between 12 and 18 h in mice and humans, respectively.

However, upon activation and during inflammation, neutrophils increase their longevity by several folds [19]. Neutrophils contain three types of granules that are sequentially formed during neutrophil differentiation [13]: primary or azurophilic granules, which contain hydrolytic enzymes such as myeloperoxidase (MPO) or neutrophil elastase (NE); secondary or specific granules which contain lactoferrin; and tertiary or gelatinase granules, which contain matrix-metalloproteinases (MMP) such as MMP-9 [20]. Neutrophils can eliminate pathogens via intracellular and extracellular mechanisms. In addition to their ability to phagocytose, generate reactive oxygen species (ROS) and release preformed granule proteins via degranulation [21], the formation of NET structures (NETosis) is yet another antimicrobial mechanism to protect the host from damage and ensure survival [22]. Among the aforementioned neutrophils defense mechanisms, NETosis is a neutrophil-specific cellular program characterized by the extracellular release of web-like DNA structures referred to as NETs [22]. NETs have an important antimicrobial role by allowing neutrophils to recognize, capture, immobilize and kill pathogens [21,22], including Gram-positive and Gram-negative bacteria, fungi, viruses and protozoan parasites. NETs are composed of extracellular chromatin fibers with a diameter of 15–17 nm where DNA and histones represent the major constituents [22,23]. These DNA fibers also include granule-derived pro-inflammatory and antimicrobial peptides, proteins and enzymes such as NE, cathepsin G, proteinase 3 (PR3), pentraxin 3 (PTX3), defensins, cathelicidin LL37 or MPO [24,25].

The process of NETosis was first described as a distinctive form of neutrophil-specific cell death [25] wherein decondensed chromatin and associated granule products are released into the extracellular space as a result of the rupture of the nuclear and plasmatic membranes [22]. Currently, two NET formation pathways have been described: (1) cell lytic NET formation or suicidal NETosis [26,27] and (2) vesicle-mediated NET formation or vital NETosis [28]. Suicidal NETosis was the first pathway to be described in 2004 [22] in terms of different stimuli such as PMA, autoantibodies or cholesterol crystals. Briefly, activated neutrophils up-regulate the glycolysis pathway inducing the activation of the extracellular signal-regulated kinase (ERK), which triggers the phosphorylation and the assembly of the NADPH oxidase complex. NADPH-oxidase regulated ROS production and increased Ca^2+^ levels, resulting in the further activation of protein-arginine deiminase 4 (PAD4).

PAD4 is a nuclear enzyme that converts arginine to citrulline on proteins, autoantibodies and on histones H3 and H4. The citrullination of histones H3 and H4 is of particular relevance for NET release: citrullination causes histones to lose much of their cationicity, thereby leading to chromatin decondensation [25]. ROS production damages the membrane of secretory granules and lysosomes and induces NE and MPO to be released from the azurophilic granules into the cytosol. Granule proteins are then translocated into the nucleus, where they promote further unfolding of the chromatin. As a consequence, the nuclear membrane breaks, and the chromatin is released into the cytosol and to the extracellular space [24].

Vital NETosis has been proposed to have a more fundamental role in regulating infection rather than sterile injury [29]. This form of NET formation can be induced through complement receptors, bacterial products, Toll-like receptor (TLR)2 and TLR4 ligands or via TLR4-activated platelets [30]. During vital NETosis, rapid NET formation occurs independently of the NADPH oxidase pathway and the chromatin fibers are expelled via vesicles without nuclear or plasma membrane disruption. DNA released during vital NETosis can also be of mitochondrial origin and occurs in a ROS-dependent mechanism [31,32,33]. Vital NETosis still allows the neutrophil to perform further functions in host response, including crawling, chemotaxis or phagocytosis. Neutrophil discrimination between its different antimicrobial responses to pathogens occurs on the basis of microbe size due to a microbe size-sensing mechanism [34]. The decision that regulates NETosis versus phagocytosis relies on the competition of the two cellular processes for the availability of NE [34]. Phagosome formation works as a checkpoint to prevent NET formation. NETosis occurs in response to specific microbes, with large microorganisms being the most effective inducers of NETosis [35]. However, pathogens have been able to develop several strategies to evade NET capture: the production and release of DNases (that trim the NET scaffold), release of virulence factor M1 (that neutralizes NET antimicrobial components activity), inhibition of the phagosome fusion, change of cell surface polarity, inhibition of NET binding by the development of polysaccharide capsules, or production of IL-10 [35,36,37,38]. Live-cell imaging is the method of choice to quantify and directly visualize the morphology, dynamics, behavior and interactions of NET formation [39,40]. A summary of the main methods and techniques used to visualize NET formation can be found in Table 1.

As previously mentioned, NETs are found under pathogenic inflammatory conditions but, interestingly, also under sterile inflammation such as autoimmune conditions, cancer or under chronic inflammatory conditions such atherosclerosis and atherothrombosis disease [62]—our main interest in this review article.

## 3. Impact of Neutrophils on Atherogenesis and Atherosclerosis Development

The pathogenesis of atherosclerosis can be divided into several key stages: (1) Endothelial dysfunction and cholesterol deposition, (2) leukocyte and SMC migration that contributes to fibrous plaque formation, (3) foam cell development and fatty streak formation, (4) intimal thickening and necrotic core formation, and (5) MMP-dependent degradation of the extracellular matrix that leads to thinning of the fibrous cap and plaque destabilization, and ultimately to plaque rupture [63]. Typically, arteries comprise three major layers [64]. The inner layer, or tunica intima, is separated from the vascular lumen through a monolayer of endothelial cells that serves as the contact surface with blood. The middle layer, or tunica media, is rich in concentric layers of vascular SMCs and embedded in a complex elastin-rich extracellular matrix. The outer layer of the vascular wall, or tunica adventitia, contains different leukocyte cell types, mast cells, fibroblasts, nerve endings and microvessels embedded in a collagen-rich matrix [64,65]. Although atherosclerosis can affect all three layers of the vascular wall, tunica intima is where atherosclerotic lesions mainly occur and develop. Atherogenesis initiates by endothelial dysfunction at sites of disturbed blood flow, such as arterial bifurcations, curvatures or branching regions [66,67]. Disturbed blood flow enhances shear stress on endothelial cells and induces modifications on endothelial gene expression characterized by an upregulation of different chemokines (CCL5, CXCL1, and CCL2) [68] and adhesion molecules (E-selectin, P-selectin, ICAM-1, VCAM-1). Such a phenotype then triggers the adhesion and infiltration of leukocytes and activated platelets into the intimal space [69,70,71]. Activated and damaged endothelium will also promote the retention of cholesterol and apolipoproteins (Apo)-containing-particles, such as low-density lipoproteins (LDLs) [66]. Subsequently, oxidative stress modifies trapped LDLs into oxidized LDL (ox-LDL) particles [72], which accelerates the inflammatory process by upregulating the formation of chemotactic gradients.

Circulating leukocytes, including lymphocytes, neutrophils and predominately monocytes, are recruited from the circulation to the site of inflammation. Particularly, macrophages differentiated from transmigrated monocytes phagocyte ox-LDL under hypercholesterolemic conditions and further differentiate into foam cells [8,66,73]. Continuous lipid uptake by foam cells will eventually promote macrophage cell death. Likewise, SMCs also undergo apoptosis in advanced lesions [11]. The excessive accumulation of inefficiently cleared apoptotic and necrotic cells, extracellular lipidic bodies and cholesterol crystals will ultimately contribute to the formation of a necrotic core and an advanced plaque [73,74]. Atheromas consist of a necrotic core surrounded by foam cells and other immune cells overall covered by a cap of SMCs -known as the fibrous cap-, lipid depositions and collagen-rich matrix [8,66]. Activated lesional macrophages release cytokines, chemokines, and MMPs. These MMPs digest extracellular matrix components and limit collagen formation, events that will thin and weaken the fibrous cap leading to plaque destabilization and rupture [74,75]. Upon plaque rupture, the pro-thrombogenic necrotic core material is released to the circulation, where it triggers platelet activation, coagulation and ultimately thrombus formation, eventually causing an acute ischemic event [66,68,74,76].

Inflammatory responses are typically governed by a first influx of abundant granulocytes that precedes a second wave of circulating monocytes recruited to the site of injury [77,78,79,80]. Yet, monocytes and macrophages are generally the most numerous myeloid cell subsets at sites of inflammation, such as in atherosclerotic lesions. In mice cremasteric venules, topical application of CCL2 can induce neutrophil and monocyte transmigration in a time-dependent manner, with a significantly higher accumulation of neutrophils within the first 2 h of the stimulus and a greater monocyte transmigration ratio at the 4-h timepoint compared to circulating neutrophil numbers [81]. In line with that, it was shown that local application of platelet-activating factor (PAF) can also induce a rapid infiltration of neutrophils in the cremaster muscle of mice just 1 h after the stimulus. Monocyte numbers, however, remained stable for the first 2 h post-PAF administration but increased significantly above baseline 4 h after the treatment. Interestingly, the influx of inflammatory monocytes was mitigated under neutropenic conditions [82].

Like neutrophils, platelets can also mediate monocyte adhesion at sites of inflammation by direct interaction with monocyte populations, which leads to the overexpression of CD40, the leukocyte receptor P-selectin glycoprotein ligand-1 (PSGL-1), CD11b or CCR2 on the monocyte surface [83,84]. Surprisingly, upon encountering endothelial injury, platelets can be activated in a matter of seconds—even before neutrophils—and are immediately recruited to the inflammatory site [85] in a well-characterized cascade of events [86]. Platelets are considered among the first cells recruited to the site of inflammation or infection and play an essential role in initiating intravascular immune responses [87].

Using a thermally induced focal liver injury, Slaba et al. were able to observe that in less than 10 min, platelets can adhere to the altered sinusoidal endothelium adjacent to the injured area, in a GPIIbIIIa (CD41)- and GPIb (CD42B)-dependent manner, and pave the way for subsequent neutrophils to enter the afflicted area, where they remain for at least 4 h [29]. Later, using a needle- and a laser-injury model, Gärtner et al. showed that platelet migration is an active, cell-autonomous and physiological process at sites of vascular injury and thrombosis and corroborated that migration takes place in a matter of seconds after the injury, which is crucial for mice host defense mechanisms [85].

It is so that only a few neutrophils are reported to be detectable in the atherosclerotic lesion [88], potentially due to their short lifespan and their ability to undergo phenotypical changes by displaying markers typically expressed on antigen-presenting cells. Yet, several studies still demonstrate the vital role of neutrophils during the early stages of atherosclerotic lesion formation [9,68,89,90,91,92,93,94,95] and advanced stages of atherosclerosis disease [11,96,97,98]. Some of the first evidence for the vital role of neutrophils in human atherosclerotic plaque development emerged after the identification of CD66b+ cells—a typical human neutrophil marker—in ruptured and eroded coronary plaques [99]. Some studies also linked high numbers of blood circulating neutrophils with the severity of atherosclerotic lesions [100], and later on, neutrophils were identified to be in close contact with intraplaque hemorrhage areas [101]. MPO-positive neutrophils were also found outnumbering present macrophages in atherosclerotic plaque regions with high inflammatory activity [102], and several neutrophil granule-derived proteins (LL-37, defensins or azurocidin) have all been identified within atherosclerotic lesions [103,104,105]. In mice, neutrophil depletion was shown to attenuate lesion formation [106,107], further supporting the role of these cells in CVD.

### 3.1. Neutrophil Migration in Atherosclerosis

Adhesion molecules, chemokines and chemokine receptors have a major role during atherogenesis by guiding neutrophil recruitment to the lesion following the classical steps of the leukocyte recruitment cascade. The activations of CX3CR1 [93,108,109] and CCR5 [94] chemokine receptors, the CCR2-CCL2 axis stimulation [95] or the overexpression of CCL2, CCL3, CCL4, CCL5, CXCL1 or CX3CL1 on endothelial cells have been reported to contribute to atherosclerosis development via neutrophil recruitment [94,95,107,108,109,110]. In particular, neutrophils can transmigrate and accumulate into the atherosclerotic lesion following different chemotactic signals released by tissue-resident macrophages or by already infiltrated leukocyte subsets, such as the leukotriene B4 (LTB4), platelet-activating factor or IL-8 expression [111,112,113,114]. Once neutrophils migrate into the lesion, they release part of their granular content [89,90] and there, several of these granule proteins bind to endothelial proteoglycans or activate endothelial cells, thus promoting further synthesis of chemoattracting cytokines.

### 3.2. Neutrophil-Mediated Immunomodulation in Atherosclerosis

Early neutrophil entry into an inflammatory site releases soluble factors and mediates a chemokine switch that induces subsequent monocyte migration due to recruitment mechanisms that involve β_2_-integrins, formyl-peptide receptors, TLR activation [115], release of granule proteins such as the LL-37 [116] or the heparin-binding protein (HBP) and several chemoattractant proteins like cathepsin G or the monocyte chemoattractant protein-1 (MCP-1) [77,117].

Neutrophils are also able to induce classical monocyte recruitment via platelet cooperation mechanisms [118]. Platelet-secreted CCL5 and neutrophil-derived cathepsin G depositions on the arterial wall can augment the endothelial adhesion of monocyte cells [89]. Human neutrophil peptide 1 (HNP1) forms a heteromer with platelet-derived CCL5 that binds to the healthy endothelium and promotes recruitment of monocytes by CCR5 binding [118]. On a different note, neutrophil-derived alpha-defensin-1, also known as HNP1, captures LDL particles in the vessel wall and contributes to lipoprotein retention [119,120]. Alarmins such as the neutrophil-derived azurocidin, also known as HBP, can also stimulate ROS formation in circulating monocytes and activated macrophages, promoting their development into foam cells, thus leading to enhanced pro-inflammatory cytokine release and phagocytic activity [121,122]. In addition, neutrophils release the monocyte-attracting granule proteins cathepsin G and cathelicidin [77,89,123], and can activate macrophages that respond by producing nitrotyrosine. The impact of this process on atherosclerosis is two-fold: (1) it significantly contributes to LDLs oxidation and (2) upregulates the expression of CD36 and CD68 on the macrophages, which are major receptors involved in the uptake of ox-LDLs [124,125].

### 3.3. Neutrophil-Mediated Vascular Dysfunction in Atherosclerosis

In an inflammatory scenario—like in the context of atherosclerosis disease—ROS production concurs with the generation of another group of redox signaling molecules: reactive nitrogen species (RNS), derived from nitric oxide. Nitric oxide is a major modulator of vascular wall inflammation and is associated with a number of intracellular effects that lead to vasorelaxation, maintenance of endothelial integrity and inhibition of aberrant leukocyte chemotaxis and excessive platelet adhesion [126]. Both ROS and RNS are crucial in the normal function and regulation of the immune response and take part in the defense mechanisms against invading pathogens [127]. Neutrophil-derived MPO is suggested to mediate and limit the bioavailability of nitric oxide close to the vascular wall, which spotlights neutrophils in the onset of endothelial dysfunction [128]. Similarly, α-defensins aggravate endothelial dysfunction by increasing radical production and the reduction of nitric oxide availability [129].

Atherosclerosis is indeed associated with reduced endothelial nitric oxide production and enhanced nitric oxide degradation due to a general endothelial dysfunction that can compromise the bioactivity of the endothelial nitric oxide synthase (eNOS) [130]. Vascular dysfunction and endothelial damage are associated with an increase in vascular oxidative stress due to an imbalance in the formation of endogenous ROS and the generation of antioxidants [131]. Excessive amounts of oxygen-derived radicals can react rapidly with nitric oxide to form peroxynitrite [132], which also contributes to the development of atherosclerotic disease by mediating lipoprotein oxidation [133]. The observation of a deficient nitric oxide-mediated vascular tone under hypercholesterolemic conditions in rabbits [134], and the lessening of atherosclerotic lesion development after the administration of L-arginine (a nitric oxide donor) in a mouse model of familial hypercholesterolaemia [135], suggest that loss of nitric oxide availability and bioactivity is a hallmark and early feature of endothelial dysfunction-associated atherogenesis and can precede progression to more advanced atherosclerotic lesions [130]. Moreover, reduced nitric oxide bioavailability corrupts its physiological anti-thrombotic role, which can result in platelet activation, aggregation and adhesion to the impaired endothelial surface [126]. Interestingly, RNS can efficiently act as inducers of NET formation by augmenting superoxide radicals and ROS formation in human neutrophils [136].

### 3.4. Neutrophil-Mediated Tissue Damage in Atherosclerosis

As previously mentioned, fibrous cap weakening due to the degradation of extracellular matrix components by MMPs leads to plaque destabilization and rupture. Neutrophil-derived MMP-8 and MMP-9 have been implicated in some of these structural changes [137,138]. MMPs degrade elastin, collagen IV and fibronectin; therefore, inhibition of these proteases resulted in decreased lesion size, reduced lesional macrophages numbers and increased collagen content within the plaque, which eventually increases plaque stability [137,138]. Neutrophils have also been implicated in superficial plaque erosion mechanisms. During plaque erosion, it has been suggested that TLR2 ligands are able to induce activation of endothelial cells and endothelial stress through mechanisms dependent on ROS production. TLR2 and neutrophils have been co-localized in common areas of the so-called erosion-prone plaques. Quillard et al. proposed a mechanism by which TLR2 stimulation induces activation of endothelial cells and instigates the recruitment of neutrophils to the arterial wall. There, neutrophils degranulate and generate oxidants that contribute to endothelial cell death and consequently to endothelial detachment and superficial erosion of the atherosclerotic lesion [10].

## 4. Impact of Neutrophil Extracellular Traps in Atherosclerosis Disease: Crosstalk of NETs with Other Cell Types

Neutrophils and NETs have been underestimated players in the development of atherosclerosis disease. Now, the recent perspective that NETs might mediate atherothrombotic events by stimulating the process of inflammation and leukocyte recruitment has expanded the original view of NETosis as a mere mechanism of defense against pathogens. NETs have been identified in both human and mice atherosclerotic lesions [139] initiated at sites of disturbed blood flow and enhanced shear stress, which are ideal conditions for neutrophil activation. NETs are recognized to be implicated in various mechanisms leading to atherogenesis, plaque erosion and plaque destabilization by their interaction and cooperative mechanisms with several other immune cell populations present in the atherosclerotic lesion (Figure 1). NETosis is now acknowledged as an emerging mechanism underlying atherogenic inflammation that likely cooperates with hyperlipidemia and oxidative stress to promote atherogenic progression and disease complications.

### 4.1. NET-Endothelial Cell Interaction

Endothelial activation and dysfunction are considered the initial spark in atherogenic inflammation. NET-induced endothelial stress and dysfunction has been suggested to be dependent on TLR2 activation and the release of interferon-α (IFN-α) in the diseased arteries [10,140]. Interestingly, reports on human atherosclerotic carotid lesions revealed the predominant presence of NETs in endothelial erosion-prone plaques compared to the classified as rupture-prone plaques [10,141]. During plaque erosion, neutrophils that localize in close contact to the inflamed endothelial surface undergo degranulation and ROS production, leading to endothelial cell death. Damaged endothelium exposes pro-thrombotic factors leading to platelet recruitment and aggregation, which further triggers neutrophil activation and NET formation. In this context, released NETs are able to induce the local activation of the alternative complement cascade (dependent on the formation of the anaphylatoxins C3a, C5a and the membrane attack complex) in the vicinity of vascular endothelial cells, promoting continued endothelial cell activation, detachment and ultimately erosion [104,142,143]. Hence, NETs seem to have a role in the development of acute atherothrombotic complications associated with the erosion-prone type of atherosclerotic lesions in a mechanism associated with induced-endothelial cell death that depends on complement activation [141].

### 4.2. NET-Monocyte Interaction

Neutrophil-derived proteins that are part of the NET backbone, such as cathepsin G or the cathelicidin-related antimicrobial peptide (CRAMP) can be deposited and transported across the inflamed endothelial surface of the atherosclerotic vessel [90], where they can trigger monocyte recruitment [82,89,144]; although it should be noted that none of these reports demonstrate whether these proteins are of NET-origin in an atherosclerotic context. Still, not only neutrophil-derived proteins, but also NET-associated histone H2a seem to be able to mediate a charge-dependent monocyte adhesion mechanism to the vasculature. This observation was shown to accelerate atherosclerosis upon lipopolysaccharide (LPS) stimulation in a murine model of acute endotoxinemia. A single dose of LPS injection in a hypercholesterolemic mice model was able to expand the size of the atherosclerotic lesions via the stimulation of NET release in the arterial lumen. It is hypothesized that NETs serve here as a platform for monocyte attachment and further accumulation, thereby fostering inflammation and potentially explaining the augmented risk for acute cardiovascular events associated with infections such as pneumonia in human patients [12]. 

### 4.3. NET-Macrophage Interaction

MPO derived from neutrophil-released NETs can bind to the macrophage mannose receptor CD206 and induce the production of ROS [70]. When activated, macrophage oxidative burst contributes to the modification of LDLs into ox-LDLs, which promotes the development of foam cells and sustains atherosclerosis progression [72]. NETs can also stimulate the macrophage scavenger receptor CD36 on these lesional macrophages and, through inflammasome activation, induce pro-inflammatory IL-1α, IL-1ß and IL-6 synthesis in primed macrophages. This process, in turn, upregulates the production of another pro-inflammatory cytokine, IL-17, from T-helper 17 cells [145]. IL-17 production is suggested to recruit neutrophils to tissue and amplify systemic inflammation [146]. Another pro-inflammatory cytokine, the macrophage-derived IL-8, seems to be elevated in serum from patients with atherosclerotic lesions and has been suggested to interact with the CXCR2 in human neutrophils, promoting NET release and thus contributing to atherosclerosis progression [147]. More recently, Josefs et al. showed that atherosclerotic NETs in diabetic mice induce a pro-inflammatory M1-like phenotype in lesional macrophages, proven by an enriched transcriptomic profile in their glycolytic and inflammasome pathways [148]. The number of NETs was augmented in diabetic mice and their interaction with macrophages contributed to the persistent macrophage-derived inflammation and impaired resolution of atherosclerotic lesions, regardless other associated risk factors. A combinatory treatment of lipid-lowering together with DNAse I injections was able to reduce macrophage and NETs numbers and decrease lesion size and instability [148]. Therefore, evidence suggests that the interplay between NETs and macrophages in a proatherogenic environment exacerbates atherosclerosis progression by inducing a pro-inflammatory phenotype in lesional macrophages.

### 4.4. NET-DCs Interaction

NET-derived DNA in combination with high levels of the antimicrobial peptide CRAMP/LL37 may influence the activation of plasmacytoid dendritic cells (pDC), identified in both murine and human atherosclerotic plaques. NET-pDCs cooperation may trigger the generation of anti-double-stranded-DNA antibodies in pDCs as well as large amounts of type I interferon (IFN-α), which facilitates the early lesion atherosclerotic development [149].

### 4.5. NET-SMCs Interaction

Neutrophils have been observed in advance atherosclerotic plaques, particularly those lesions characterized by a large lipid core, high macrophage numbers, low SMC numbers and little collagen depositions [96], which points to neutrophils as key players in the plaque destabilization process. Mechanistically, recruited neutrophils to the vascular lumen are able to physically interact with fibrous cap resident SMCs, which can trigger further activation of neutrophils. SMC-synthesized C–C motif chemokine ligand 7 (CCL7) enhances NET formation, which subsequently prompts lytic cell death of SMCs. In agreement with earlier reports on NET-derived histones displaying cytotoxic activity [150], the study shows that histone H4 present in NETs seems to be the key mediator in the induction of arterial wall SMC cell death through the formation of charge-dependent cell membrane pores and membrane destabilization [11]. SMC depletion leads to thinning of the fibrous cap by removing a source of interstitial collagen synthesis hence leading to plaque instability. In agreement, neutrophilic mice present higher numbers of SMC death while mice depleted of neutrophils or mice lacking PAD4 (unable to produce NETs) exhibited atherosclerotic lesions with an increased SMC content and fewer characteristics of advanced plaque vulnerability [11]. This suggests a neutrophil role in regulating SMC viability in atherosclerosis and a direct cytotoxic capacity of NETs. 

### 4.6. NET-Platelet Interaction

Some NET components display pro-thrombotic competencies and can instigate the coagulation cascade initiation via thrombin activation [151]. NETs can form a fibrin-like backbone and pave the way for platelet adhesion, activation and aggregation [151,152]. In fact, increased NET formation has been associated with hypercoagulability in septic patients as well as in patients with chronic vascular disorders [153]. Particularly, NET-derived histones H3 and H4 have been reported to induce platelet activation and aggregation by, among others, recruiting fibrinogen [154] in a TLR2- and TLR4-dependent mechanism [155] that eventually accelerates thrombin production [151]. NE and cathepsin G have been shown to enhance tissue factor- and factor XII-driven coagulation via proteolysis of the tissue factor pathway inhibitor (TFPI) [156].

Moreover, NETs have been suggested to participate in thrombi formation, growth and stabilization by providing a scaffold for fibrin deposition, red blood cells and platelet aggregation within the thrombus, and by enhancing the accumulation of pro-coagulant and pro-thrombotic molecules such as von Willebrand factor (VWF), fibronectin, fibrinogen [151], factor XII [157] and tissue factor [158]. As such, the histone-DNA backbone of NETs is thought to add stability to the fibrin scaffold in thrombi [159]. In turn, activated platelets can also induce NET formation both through mechanisms involving TLR4 [30], high mobility group box 1 (HMGB1) [160], and P-selectin [161], showing a complex crosstalk between platelets, activated neutrophils and NETosis. For example, the evaluation of thrombectomy samples from infarcted patients revealed that at plaque rupture-prone areas, platelet-derived HMGB1—a nuclear protein that can function as a damage-associated molecular pattern (DAMP)—can induce NET formation and promote delivery of tissue factor, which would spotlight NETs as main contributors of plaque rupture and thrombus formation events [160]. LPS-stimulated platelets have been shown to promote NET formation through TLR4 activity in the liver sinusoids [30]. P-selectin CD62P on the platelet surface binds to PSGL-1 on the neutrophil surface and is also able to mediate NET release [162]. Platelet-derived chemokines—such as CCL3 or the CCL5/CXCL4 heterodimer—instructs the recruitment of immune cells to the site of inflammation and drive neutrophils to form NET structures [163].

Neutrophils can be directly activated by heparin/platelet factor 4/IgG antibody complexes through their FcγRIIa receptors, which also trigger NET formation [164]. Moreover, tissue factor and platelet-derived VWF via the platelet GPIbα are implicated in NET-mediated deep vein thrombosis in mice [157,161]. The interplay of NETs, activated platelets and clotting factors revolve around a positive feedback loop that coordinates a pro-thrombotic cycle of coagulation and pro-inflammatory response that could ultimately occlude the vessel and induce organ damage. In sum, enough evidence supports that NETs have a decisive role not only in atherogenesis and lesion progression, but also in atherothrombotic events formation.

## 5. The Interplay of Neutrophil Extracellular Traps with Established Risk Factors for Cardiovascular Disease

Lifestyle-derived risk factors are gaining attention in recent years as environmental modifiers of the inflammatory process as well as of CVDs that manifest with chronic, overactive and dysfunctional inflammation, such as atherosclerosis [165]. Unbalanced dietary habits, physical inactivity, stress, smoking and lack of sleep quality impact and aggravate cardiovascular inflammation and enhance the risk of associated diseases. This occurs, in part, by modifying the niche of neutrophil production as well as neutrophil activation and functions [166]. In the next section, we review the evidence that brings together several lifestyle factors and their interplay with one of the neutrophil effector functions, NETosis, within a particular, but not exclusive, context of atherosclerosis (Figure 2).

### 5.1. NETs & Obesity

Obesity is a multifactorial disease that results from complex interactions between genetic and environmental factors [167,168]. Obesity is characterized by an excessive accumulation of white adipose tissue (WAT) generated as a consequence of a disbalance between excessive nutrient intake and deficient energy expenditure [169]. Obesity is also associated with a general systemic and chronic low-grade inflammation [170] and is frequently linked with other chronic complications (hyperglycemia, hyperlipidemia and hypertension) that ultimately contribute to cardiovascular and metabolic morbidity and mortality [171,172,173] (Figure 2a). In an obese context, oxygen supply to hypertrophied adipocytes seems to be uneven. Hypoxic environment induces necrosis, macrophage infiltration [174] and the overproduction of pro-inflammatory cytokines (TNF-α, IL-6, IL-8) in WAT [172,175] that reinforce the obese-associated inflammatory scenario [176]. Neutrophilia and increased neutrophil activity are reported in hyperlipidemic obese subjects [177,178].

Indeed, Moorthy et al. showed that the neutrophils of high-fat diet (HFD)-fed mice are more prone to spontaneous NET formation in the lungs in comparison to neutrophils derived from low-fat diet-fed animals [179]. Both neutrophilia and increased neutrophil activity have been linked to an increased pro-inflammatory cytokine cascade from adipocytes and correlate with high blood levels of LDL particles [1,180]. Recent studies have revealed that feeding mice a HFD induces neutrophil recruitment to adipose tissue and increases NET formation due to higher oxidative stress levels [179,181], even in the absence of other risk factors or other cardiometabolic complications [182,183]. Increased neutrophil recruitment to adipose tissue is maintained in the chronic obese state, constituting yet another feature of the pro-inflammatory response in obesity [184,185]. D’Abbondanza et al. found a correlation between plasmatic MPO-DNA complex levels—as an indirect measure of NET formation—and some anthropometric and glycometabolic parameters. NET levels were reported higher in morbid obese patients compared to healthy controls and their numbers were associated with a higher number of thromboembolic events [186].

More recently, however, Cichon et al. observed that in the context of general sepsis, neutrophils from obese mice seem to have a reduced NET formation capacity compared to normal-fed animals [187]. Of interest, purified neutrophils from obese patients seemed to have a reduced capacity to generate NETs in vitro, which could suggest functional exhaustion from excessive NET formation while in circulation [188]. Thus, the role of NETs in experimental models of obesity and in obese patients seems to be controversial. For example, in a mouse model of HFD-induced obesity, Braster et al. were unable to find differences in metabolic or in inflammatory parameters in WAT and livers from pharmacologically (Cl-amidine injected) PAD4-deficient mice, compared to obese-controlled animals. The authors concluded that targeting NETs may not have any clinical relevance to the onset of obesity, but does not rule out a potential therapeutic role in advanced obesity and its associated pathologies [189].

On the other hand, Wang et al. showed that pharmacological intervention with NET formation (via Cl-Amidine or DNAse administration) mitigates the endothelial dysfunction, associated atherosclerosis and the endothelium-dependent vasodilation in mesenteric arteries that occurs in the HFD-induced obese mice models. They suggest that an abnormal production of NET-derived MPO would trigger the generation of ROS in close proximity to the vascular wall, which puts the spotlight on NETs as key players in obesity-related endothelial damage [181]. Accordingly, the role of NETs in obesity as well as their input to atherogenesis in an obese context remains, up to now, controversial. Disparities can be partially explained by the usage of different animal models as well as clinical studies made in non-comparable patient groups, so it would need further and deeper evaluation.

### 5.2. NETs & Hyperlipidemia

Elevated levels of blood lipids such as cholesterol and high concentrations of serum triglycerides (TG) are well-established risk factors for CVD (Figure 2b). Indeed, the deposition of cholesterol in the arterial wall is considered one of the key initiating events of atherogenesis [190]. Later, excessive circulating cholesterol can precipitate into monohydrate cholesterol crystals and accumulate in the vessel wall, where it contributes to the formation of advanced atherosclerotic plaques and eventually to plaque rupture [191,192]. Interestingly, hypercholesterolemia triggers monocytosis and neutrophilia due to enhanced granulopoiesis and myeloid cell mobilization from the bone marrow into the circulation [107].

In addition, neutrophil counts seem to be postprandially elevated in healthy subjects when TG levels rise. Postprandial lipidemia and neutrophilia have been associated with, interestingly, a number of inflammatory hallmarks of atherosclerosis: (a) the upregulation of leukocyte activation markers CD11b and CD66b, (b) the production of pro-inflammatory cytokines, (c) oxidative stress, and (d) have been suggested to contribute to endothelial dysfunction [193]. Moreover, total leukocyte counts and, in particular, neutrophil counts were positively associated with several components of the Metabolic Syndrome, among them, hypertriglyceridemia and low HDL cholesterol in a three-year follow-up study [194]. 

More specifically, NET levels in atherosclerotic plaques and in blood plasma seem to be consistently increased under HFD [11,195]. In this regard, Warnatsch et al. demonstrated the consequences of targeting NETs in the pathogenesis of atherosclerosis by observing diminished atherosclerotic lesion size, lower systemic IL-1β levels and decreased NETosis ratios in HFD-fed, Apo-E-deficient and LDL-deficient mice when they lack the expression of the neutrophil proteases NE or PR3 [145]. Such evidence supports the notion that hyperlipidemic conditions are associated with an increased capacity of NETosis, which could impact the patient’s inflammatory status, particularly in an atherosclerotic context. In a non-atherosclerotic context, Jeremic et al. showed that cholesterol crystals injected in mice air pouches induced tophus-like clumps composed of aggregated NETs displaying markers that have also been identified in the necrotic core areas of atherosclerotic plaques [196].

In atherosclerosis, again, it seems that cholesterol crystals can particularly function as sterile danger signals. For example, in aortic roots of mice, cholesterol crystals are demonstrated to trigger neutrophils to release ROS-dependent NETs by translocating NE and PR3 into the nucleus [145]. In vivo, cholesterol crystal-triggered NETs can prime lesional macrophages into releasing inflammatory cytokines such as IL-1ß and IL-6, which will further activate Th-17 cells that amplify the immune cell recruitment cascade into the atherosclerotic lesions [145]. In vitro, however, the depletion of cholesterol by administration of statins or the methyl-β-cyclodextrin (MβCD)—an inhibitor of the cholesterol domains in the cell surface—results in an increased NET formation capacity in primary neutrophils [197,198]. Earlier, Duewell et al. suggested that the uptake of cholesterol crystals by lysosomes induces the NOD-like receptor protein 3 (NLRP3) inflammasome activation [199], which was later suggested to cause gasdermin-D dependent pore formation in the plasma membrane of macrophages and neutrophils, inducing NET release and contributing to plaque erosion and thrombosis [200]. In this regard, Westerterp et al. found that the accumulation of cholesterol in myeloid cells activates the NLRP3 inflammasome in macrophages and triggers neutrophil accumulation and NET formation in mice atherosclerotic plaques. Patients with increased myeloid cholesterol content display signs of inflammasome activation, suggesting human relevance in the potential cholesterol-NLRP3-NET axis described earlier [201].

### 5.3. NETs & Hypertension

Arterial hypertension has long been considered a leading risk factor for cardiovascular disease. Interestingly, high neutrophil blood counts and elevated neutrophil-to-lymphocyte ratio are considered clinical hallmarks of hypertension disease development [202,203]. Moreover, in addition to elevated neutrophil counts, hypertensive patients also show unleashed neutrophil responses that can modulate the microenvironment in blood vessels. Neutrophils could, in turn, mediate vascular damage and promote endothelial cell death by secreting ROS, which could ultimately make them active participants in the development of arterial hypertension [204].

Earlier in 2017, Hofbauer et al. observed a positive correlation between the NETosis capacity in isolated neutrophils from coronary artery disease (CAD) patients and their levels of blood pressure. This ROS-dependent NET mechanism seems to be mediated by the angiotensin-II receptor [205]. More recently, Li et al. evaluated the plasma levels of cf-DNA and MPO-DNA complexes -as an indirect measure of NET activity- in patients that suffered from primary hypertension and hyperhomocysteinemia or primary hypertension alone. Hypertensive patients display elevated levels of NET release compared to controls. Hypertensive and hyperhomocysteinemic patients showed the highest levels of NET release [206]. High levels of homocysteine in blood is a condition that increases thrombotic events and is a known risk factor for cardiovascular events due to endothelial damage and increased oxidative stress [207]. Hence, the authors of the study suggest that hypertension and high homocysteine may be a trigger for NET release, predisposing patients to a more hypercoagulable state and a higher thrombotic risk [206].

On another note, Korai et al. demonstrated that targeting NETs might be a potential therapeutic strategy to prevent aneurysmal rupture in murine models of intracranial aneurysms. The authors suggest that this NET formation may be triggered by endothelial damage, hemodynamic stress and the general context of inflammation that exist in the aneurysm vascular wall, conditions that resemble those found in atherosclerosis. Knowing that high blood pressure has been shown to play an important role in the formation and rupture of intracranial aneurysms, this is another example of the interplay between NETs, hypertension and hypertension-associated conditions [208]. Briefly, under pathological conditions like arterial hypertension there seems to exist an imbalance towards vascular ROS release against the generation of antioxidant molecules, which would ultimately augment vascular damage and endothelial dysfunction. This would provide the ideal framework for neutrophils to secrete pro-inflammatory mediators and increase their NETosis capacity. In turn, when activated, neutrophils could cause further arterial damage and so further increase blood pressure levels. That way, hypertension could predispose to and accelerate atherosclerosis, being neutrophils and NETs main players in the scenario. 

### 5.4. NETs & Physical Activity

Sedentarism -lack of regular physical activity- impacts on the whole immune system functionality and increases atherosclerotic risk development [209]. In fact, physical inactivity is considered one of the leading causes of the higher incidence of metabolic and cardiovascular diseases and premature deaths worldwide [210,211,212]. Exercise improves low-grade systemic inflammation, promotes anti-inflammatory pathways (reduces IL-6 and TNF-α mRNA levels) [213,214], induces higher systemic levels of antioxidant enzymes, seems to be associated with lower levels of circulating neutrophils and monocytes both in humans and mice [215,216,217], and reduces granulocytes capacity to produce pro-inflammatory cytokines (IL-1β, IL-6, IL-7, CRP and TNF) [218,219].

Interestingly, neutrophil counts also appeared to be reduced in blood and peripheral organs in physically active individuals that have chronic inflammatory conditions compared to sedentary subjects [220]. It has been shown that after an acute bout of exhausting exercise, the levels of lactic acid (LA) can increase up to 20-fold in blood plasma [221]. Shi et al. demonstrated a negative regulation by LA in neutrophils netting capacity. C57BL/6 mice were submitted to 145 min of exhaustive running, after which concentrations of LA and cf-DNA levels increased in plasma, while MPO-DNA complex levels -a distinctive indicator of NET release- were decreased. In vitro, increasing LA concentrations were also associated with decreased NET release ratios and reduced ROS formation [222]. It was also that found that physical inactivity induces vascular NADPH oxidase expression and higher vascular superoxide levels in mice, promoting endothelial dysfunction and atherogenesis progression [223]. Thus, one could hypothesize that in the context of atherosclerosis, circulating neutrophils from physically active subjects could potentially be less reactive and less prone to undergo NETosis. As such, “exercised” neutrophils would inflict less damage to the vascular endothelium. Altogether, these ideas would support the role of exercise as an anti-inflammatory practice able to regulate neutrophil effector functions.

### 5.5. NETs & Sleep Health

Poor and insufficient sleep is recognized as a significant risk factor for immune and cardiometabolic diseases [224,225]. Evidence suggests that the alteration of healthy sleeping patterns and disruption of the normal circadian rhythms influence leukocyte and lipid supply in the circulation and ultimately favors atherogenesis [226]. Shorter sleeping times and sleep fragmentation in humans raise neutrophil and monocyte cell counts [226,227] in blood and peripheral tissues [228]. Sleep fragmentation also increases inflammatory markers (IL-1, IL-6, IL-17, CRP and TNF-α) [225] and triggers a low-grade chronic inflammatory status [229], hereby increasing atherosclerosis risk and severity [230], even independently of other atherosclerotic risk factors [227]. Experimentally induced sleep fragmentation in mice results in larger atherosclerotic lesions, which is directly associated with an increase in neutrophils and monocyte blood levels [231,232]. In addition, Christoffersson et al. showed that following a night of total sleep deprivation (TSD), the blood circulating neutrophil population redistributed in the next morning and displayed distinct mean fluorescence intensity (MFI) levels of CXCR2, CD62L and CD16 markers compared to subjects that had 8 h of regular sleep. Following TSD, circulating neutrophils also displayed increased levels of the neutrophil activation marker CD11b [233]. This may point to an enhanced pro-inflammatory state of peripheral blood neutrophils that could potentially explain the increased risk of developing CVDs in individuals with poor chronic poor sleeping patterns. Interestingly, Adrover et al. showed that young neutrophils (CXCR2^high^, CD62L^high^) freshly released into circulation display a higher granule content and augmented NET-forming capacity compared to aged PMNs (CXCR2^low^, CXCR4^high^) under homeostatic conditions [234]. 

Conversely, Zhang et al. showed that under septic conditions, aged neutrophils in mice are the ones exhibiting enhanced NET formation [235]. Unfortunately, no current scientific reports dive into the mechanisms linking NETs, sleep deprivation and atherosclerosis. However, due to the highly recognized role of neutrophils and NETs in atherosclerosis progression, one could think that the augmented number of highly activated neutrophils reported during sleep altered patterns could display elevated NETosis ratios that would contribute to endothelial damage progression and ultimately foster an atherosclerotic-prone scenario in the arterial wall. Consequently, Said et al. observed that sleep deprivation in humans leads to a decreased neutrophil phagocytosis capacity and a reduced activation of the NADPH oxidase pathway [236]. Since it is known that phagocytosis and NETosis are regulated neutrophil functions that rely on the competition and availability of NE [34], it could make sense to think that sleep deprivation would not only decrease the neutrophil phagocytic activity but rather increase their netting capacity under sleep fragmentation conditions. 

### 5.6. NETs & Smoking

Tobacco smoking is associated with a wide range of systematic pathologies and have long been considered a major risk factor for the development of atherosclerotic disease (Figure 2c). Clinical as well as animal studies have shown that cigarette smoking promotes endothelial dysfunction and accelerates atherosclerosis [237,238] by, for example, enlarging atheroma-covered areas in the arterial wall of cholesterol fed-rabbits [239]. In rats, exposure of aortas to cigarette smoke extracts results in severe endothelial dysfunction [240]. It was suggested that cigarette smoke impairs NO-mediated endothelial functions and elicits cardiovascular toxicity through a mechanism that depends on the generation of superoxide anions (O_2_^−^) and ROS [241,242]. Clinical studies report that smokers also have an increased expression of pro-inflammatory markers including IL-6, CRP, E-selectin, P-selectin, and intercellular adhesion molecule-1 [243], all of them well-known players in the pathogenesis of atherosclerosis. 

Currently, it is well-known that smoking elicits a state of oxidative stress in endothelial cells, and recently it has been suggested that smoking can also impact immune cell populations functionalities. In fact, exposure of murine and human neutrophils to chronic tobacco smoke was shown to induce abundant NET formation [244,245], and neutrophils isolated from smoking subjects display elevated ratios of spontaneous and PMA-induced NETosis [246]. Hosseinzadeh et al. demonstrated that isolated human neutrophils are able to undergo NETosis via the stimulation of the nicotinic acetylcholine receptor (nAChR) in response to nicotine stimulation, thus potentially exacerbating inflammatory pathways and inducing tissue damage [247]. As tobacco smoking a major risk factor for CVD, one could speculate that the smoke-induced neutrophil activation and NETosis pathways could also have a potential role in the vascular damage associated with atherosclerosis and contribute to disease progression.

## 6. Therapeutical Approach

Based on its critical role on atherogenesis and atherosclerosis disease progression, NETs are nowadays looked upon as novel targets for treatment and prevention of atherosclerotic disease and plaque complications. The administration of the general PAD inhibitor Cl-Amidine in mice prevents NET formation in the plaque. Abolished NET formation results in a decline of the number of intimal macrophages, decreased neutrophil recruitment to the arterial wall, decrease arterial IFN-α levels and ultimately a significant reduction in the atherosclerotic lesion size [140]. Generally, mice either genetically or pharmacologically deficient in NET formation (PAD4- or NE-targeted) are able to decrease the expression of pro-inflammatory cytokines in the aortic region, have reduced and more stable plaques in the aortic root, are more protected from the HFD-induced adipocyte inflammation and show improved metabolic indices (i.e., improved glucose tolerance and insulin resistance) [195,248].

Recently, Molinaro et al. used a nanoparticle-delivery approach to deliver the PAD4 inhibitor GSK484 to collagen-IV rich areas of the endothelium that shows features of superficial erosion. The systemic administration of these targeted nanoparticles limits NETs formation by inhibiting PAD4 enzymatic activity and preserves endothelial integrity [91]. All in all, PAD4 is a proven therapeutical target by which NET-induced atherosclerosis can be significantly improved. DNAse I treatment has also been suggested as a beneficial therapeutic tool in neutrophil-derived diseases in which NET formation plays a pathogenic role: Warnatsch et al. demonstrated that systemic DNAse I injection reduced lesion size, amount of lesional NETs as well as amount of pro-inflammatory cytokines in murine atherosclerotic plaques of ApoE/PR3/NE^−/−^ animals [145]. However, similar studies failed to corroborate these observations in the same murine model during early stages of atherosclerosis [92]. In line with that, Franck et al. showed that DNAse I administration in Ldlr/PAD4-deficient murine models resulted in a decreased superficial erosion of the endothelium, limiting the number of adherent neutrophils and increasing the survival of endothelial cells [141]. Still, some studies led to a controversial idea regarding the systemic administration of DNAse I as it has also been shown to accelerate tissue-type plasminogen activator (tPA)-mediated thrombolysis in human coronary disease; based on the idea that NETs have a clot-stabilizing effect in thrombi formation [249].

Neutralizing antibodies to histone H4 or histone H2a could also be therapeutically useful as a NET-mediating strategy that has been shown to improve plaque stability in mice models of atherosclerosis [11,12]. Other validated therapeutic approaches that block the release of NETs include NET-resident proteins neutralization, complement inhibitors, ROS scavenging, low molecular weight heparin administration [250], phosphodiesterase 4 (PDE4) usage, or partial removal of neutrophils. All things considered, anti-NET therapy postulates as an advantageous treatment in atherosclerosis disease due to its low relative immunosuppression and low compromise with the patient’s immune defense.

### NETs as Biomarkers of Cardiovascular Disease

Recent studies indicate that NET components can serve as CVD biomarkers, and their evaluation can be used as a diagnostic tool to predict the severity of atherosclerosis disease and the risk of future cardiovascular events. The presence in blood of double-stranded extracellular DNA, DNA-MPO complexes, citrullinated histone H4 (citH4) and citH3, cf-DNA, nucleosomes or NE, among other NET components, has been embraced as different ways to evidence the participation of NETosis in atherosclerosis progression and disease [251].

Borissoff et al. were able to demonstrate an independent association between NET formation, coronary artery disease and the development of a prothrombotic state by identifying that elevated levels in plasma of double-stranded DNA, nucleosomes and MPO-DNA complexes positively correlated with thrombin generation and the occurrence of a major adverse cardiovascular event in patients [252]. Erosion-prone classified atherosclerotic plaques, unlike rupture-prone plaques, display colocalization of the neutrophil marker CD66b with different NET formation markers such as citH4 and NE [141]. More recently, Vallés et al. were able to identify elevated levels of several markers of NET formation (cf-DNA, nucleosomes, and citH3) in the blood plasma of patients that underwent an acute ischemic stroke when compared to healthy subjects, particularly those patients older than 65 years and with a history of atrial fibrillation (AF) and severe stroke scores [253]. Yet, and despite extracellular DNA being considered the hallmark of NET formation, it remains complicated to elucidate its cellular origin and determine whether free DNA truly derives from active NETosis or is a remnant of apoptosis, necrosis or purely cell debris [254]. Some other specific examples of NET-derived biomarkers in cardiovascular disease were extensively reviewed elsewhere [255,256].

## 7. Conclusions

For many years, despite neutrophils being the most abundant leukocyte in the bloodstream in humans, and the principal responders during acute inflammatory conditions, these cells have been underestimated players in the context of cardiovascular inflammation and atherosclerosis disease. Now neutrophils and our particular interest in this review article, NETs, have revealed to have key functions and impact all stages of atherosclerosis and atherothrombosis disease, and so they have emerged as potential therapeutical targets in the context of cardiovascular inflammation.

In vascular tissues, the chronic inflammatory context that is a common hallmark in conditions such as atherosclerosis, entails the trigger of the innate immune response and the activation of mobilized granulocytes. Upon activation, neutrophils can undergo NET release and sustain inflammation by communicating with different cell subsets present in the atherosclerotic lesion. There, NET formation seems to lead to a vicious cycle of endothelial tissue activation and injury, pro-inflammatory cytokine release, further neutrophil recruitment and recurrent NET production, which ultimately results in a condition of sustained chronic inflammation, cellular dysfunction and tissue damage. Accordingly, NETs are revealed as unusual suspects in the link between inflammation, innate immunity, oxidative stress, endothelial dysfunction, arterial wall inflammation and cardiovascular disease.

Lifestyle-derived factors highly impact the risk of developing CVD and influence novel disease-associated pathways that involve inflammation, oxidative stress and tissue and organ dysfunction. Adherence to a healthy lifestyle by improving diet and sleeping habits and by including modest alterations of lifestyle routines such as regular physical activity, can have substantial positive effects on cardiovascular risk. A healthy lifestyle can unequivocally help reduce systemic inflammation, lowering disease burden and decreasing the risk of mortality among the global population. Accordingly, it would make sense to properly unravel the mechanism of action of all the aforementioned lifestyle-derived risk factors along with their interactions with inflammation, neutrophils and NETs, and tackle them by using preventive and therapeutic strategies that would reduce disease complications and burden, and would ultimately improve human health.

## Figures and Tables

**Figure 1 cells-10-01985-f001:**
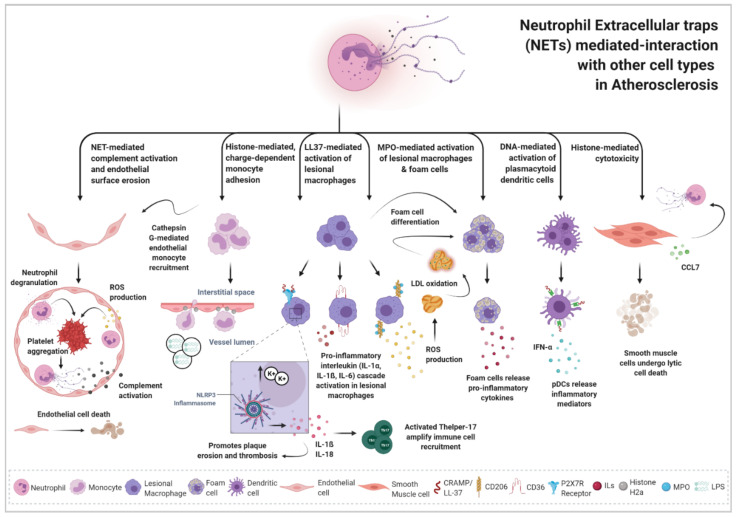
Neutrophil extracellular traps-mediated interaction with different cell populations in atherosclerosis. The highly activated neutrophil during atherogenesis—after the lesion initiates at sites of disturbed blood flow and enhanced shear stress—finds there the ideal inflammatory-prone scenario to undergo NETosis. Neutrophil extracellular traps (NETs) are able to mediate different responses in the diverse immune and structural cell populations that co-exist within atherosclerotic lesions and so they can drive lesion progression. NETs sustain a chronic inflammatory recruitment of monocytes, monocyte-derived macrophages and lymphocytes by either paving the way for monocyte recruitment through histone-mediated and charge-dependent monocyte adhesion mechanisms or by triggering the NLRP3 inflammasome activation in macrophages, which through the release of the pro-inflammatory cytokine IL-1ß, can activate T-helper 17 lymphocytes and amplify immune cell recruitment in the artery wall. NETs can also cast tissue damage in the arterial wall towards endothelial and smooth muscle cells due to some recently described NET-mediated cytotoxicity mechanisms. Through complement activation, NETs induce endothelial erosion in the tunica intima, and through a histone-mediated cytotoxicity mechanism, NETs can directly mediate the apoptosis of smooth muscle cells in the tunica intima and the tunica media. Moreover, NET components such as the myeloperoxidase (MPO) granule protein, or the cathelicidin-related antimicrobial peptide (CRAMP)—LL-37 in humans—can bind to different macrophage receptors, such as the macrophage mannose receptor CD206 or the macrophage scavenger receptor CD36 and increase oxidative stress. Reactive oxygen species (ROS) production triggers the oxidation of deposited low-density lipoprotein (LDL) and cholesterol which, in turn, promotes further NETosis, triggers foam cell differentiation in lesional macrophages and ultimately contributes to advance lesion progression and development. Created with BioRender.com.

**Figure 2 cells-10-01985-f002:**
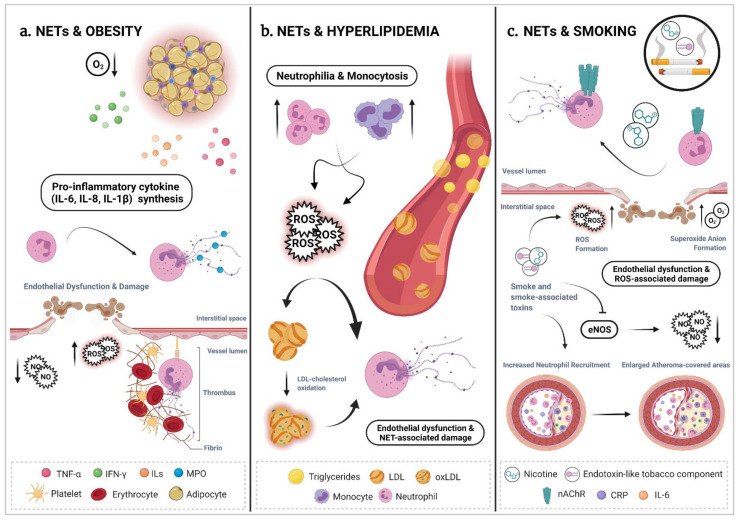
Proposed neutrophil extracellular traps mechanisms of action associated to some cardiovascular and lifestyle-derived risk factors. (**a**) In an obese context, excessive accumulation of white adipose tissue (WAT) appears to be depleted of physiological oxygen levels and is associated to a systemic and chronic low-grade inflammatory profile. Hypoxia in WAT induces necrosis, immune cell infiltration, oxidative stress and a general overproduction of pro-inflammatory cytokines. This scenario facilitates “obese” neutrophils to undergo spontaneous NETosis. Augmented neutrophil extracellular traps (NET) levels have also been associated to an increased number of thromboembolic events in obese patients due to the NET-mediated endothelial damage and dysfunction. (**b**) Hyperlipidemia reflects a status of elevated levels of lipids —such as cholesterol and triglycerides— in the blood. Hypercholesterolemia triggers monocytosis and neutrophilia, conditions that are associated with the release of pro-inflammatory cytokines and the production of oxidative stress. Deposited cholesterol crystals and oxidized LDL particles can function as sterile danger signals that induce the release of NETs, contributing so to endothelial dysfunction and NET-associated damage. Cholesterol crystal-triggered NETs can prompt the NLRP3 inflammasome activation in primed lesional macrophages, and arouse them into releasing inflammatory cytokines such as IL-1ß and IL-18, which will further amplify immune cell recruitment, and, in turn, increment NETosis ratios. (**c**) Cigarette smoke impairs nitric oxide-mediated endothelial function and elicits cardiovascular toxicity due to augmented ROS and superoxide anions (O_2_^−^) generation. Smoker subjects display elevated levels of pro-inflammatory markers such as IL-6 or CRP, both recognized players in atherosclerosis. NETs have been proposed now to play a main role in this context. Neutrophils are able to undergo NETosis via the stimulation of the nicotinic acetylcholine receptor in response to nicotine stimulation. Cigarette smoke, nicotine, and cigarette-associated toxins are able to exacerbate inflammatory pathways, induce tissue damage and enlarge atheroma-covered areas in the arterial wall in an atherosclerotic context, all due to an increased NET formation capacity. Figure legend: ROS (reactive oxygen species), NO (nitric oxide), nAChR (nicotinic acetylcholine receptor), eNOS (endothelial nitric oxide synthase), CRP (C-reactive protein). Created with BioRender.com.

**Table 1 cells-10-01985-t001:** NET Visualization Approaches.

Identified Target	Technique	Methodology	Dye	Sample	Strengths	Weaknesses	Ref.
NETs	Spinning disk confocal microscopy	Live-cell imaging to visualize morphology, dynamics, behavior of NETs and interactions between NET formation and live pathogens	Fluorescent Bacteria and NET markers	Animal Model	Evaluation of NET formation in situ and in vivo Allows NET dynamics evaluation and kinetic studies	Invasive technique that can induce unwanted inflammatory responses Poor temporal resolution Cardiac and Respiratory movement might compromise imaging Expensive, costly, time-consuming and technically challenging Long acquisition times	[41,42].
NETs	Multi-/ Two-photon microscopy	Live-cell imaging to visualize morphology, dynamics, behavior of NETs and interactions between NET formation and live pathogens	Fluorescent Bacteria and NET markers	Animal Model	Evaluation of NET formation in situ and in vivo Allows NET dynamics evaluation and kinetic studies	Invasive technique that can induce unwanted inflammatory responses Poor temporal resolution Cardiac and respiratory movement might compromise imaging Expensive, costly, time-consuming and technically challenging Long acquisition times	[19,42,43,44].
NETs	Intravital Microscopy	Live-cell imaging to visualize morphology, dynamics, behavior of NETs and interactions between NET formation and live pathogens	Fluorescent Bacteria and NET markers	Animal Model	Evaluation of NET formation in situ and in vivo Allows NET dynamics evaluation and kinetic studies	Invasive Technique that can induce unwanted inflammatory responses Poor temporal resolution Cardiac and Respiratory movement might compromise imaging Expensive, costly, time-consuming and technically challenging Long acquisition times	[41,42].
Soluble NET remnants cf-DNA	Fluorescent Reader		PicoGreen^®^	Plasma Serum Fluids	Objective Quantitative Can potentially detect in vivo NET formation	Low Specificity Cell-free DNA can reflect lytic cell death mechanisms (Necrosis)	[45,46].
DNA-MPO Complexes	ELISA			Plasma Serum Fluids	Objective Quantitative	Low Specificity MPO (high-cationic nature) can bind to negatively charged cf-DNA MPO can reflect on neutrophil and macrophage activation not related to NET release	[47,48,49,50]
DNA-NE Complexes	ELISA			Plasma Serum Fluids	Objective Quantitative	NE can reflect on neutrophil activation and degranulation not related to NET release	[47,48,50].
Citrullinated Histone H3	Refined ELISA			Human Plasma	Objective Quantitative Citrullination is the gold hallmark of NET formation	No consensus on a standard ELISA assay to monitor NETosis	[51,52].
MPO citH3	Flow Cytometry	Detection of NET components attached to the neutrophil cell surface	Fluorescent antibodies against MPO and citrullinated histones	Primary cells Cell lines	Objective Unbiased Automated Can be combined with cell-sorting techniques	Does not detect citH3-independent events Potentially able to report live NETosis events but misses lysed cells that underwent NETosis previously	[46,53].
DNA backbone	IF	Co-localization of extracellular DNA, neutrophil markers, neutrophil-derived granule proteins and modified histones	DNA-intercalating dyes (DAPI, PI, SYTOX^®^ Green)	Tissue Sections	Can differentiate between necrosis and NETosis	Biased selection of the field of view might affect results Clump of NETs derived from multiple cells count as a single event	[26,54,55,56,57,58,59,60,61]
Neutrophils	IF	Co-localization of extracellular DNA, neutrophil markers, neutrophil-derived granule proteins and modified histones	Fluorescent antibodies against Ly6G (in mice) and CD66b, NE or CD177 (in humans)	Tissue Sections	Can differentiate between necrosis and NETosis	Biased selection of the field of view might affect results Clump of NETs derived from multiple cells count as a single event	[26,54,55,56,57,58,59,60,61]
Neutrophilderived Granule proteins	IF	Co-localization of extracellular DNA, neutrophil markers, neutrophil-derived granule proteins and modified histones	Fluorescent antibodies against MPO, NE or LL37	Tissue Sections	Can differentiate between necrosis and NETosis	Biased selection of the field of view might affect results Clump of NETs derived from multiple cells count as a single event	[26,54,55,56,57,58,59,60,61]
Citrullinated Histone	IF	Co-localization of extracellular DNA, neutrophil markers, neutrophil-derived granule proteins and modified histones	Fluorescent antibodies against citH3, citH4	Tissue Sections	Can differentiate between necrosis and NETosis	Biased selection of the field of view might affect results Clump of NETs derived from multiple cells count as a single event	[26,54,55,56,57,58,59,60,61]
NETs	IF	NET detection reagents that are based on fluorescent, chromatin-binding polymers	PlaNET reagents	Primary cells	High Specificity Stain NETs only in activated cells Undetectable in necrotic cells	Non-specific signals reported Not suitable for NETosis kinetic studies	[50]

For a far more detailed approach and understanding of NET visualization techniques, we refer the reader to another excellent review article [39]. cf-DNA, cell free-DNA; MPO, Myeloperoxidase; ELISA, enzyme-linked immunosorbent assay NE, Neutrophil Elastase; IF, Immunofluorescence; PI, Propidium Iodide; citH3, citrullinated Histone3.

## Data Availability

No new data was created or analyzed in this study. Data sharing is not applicable to this article.

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
