# Peer review of "Contemporary Lifestyle and Neutrophil Extracellular Traps: An Emerging Link in Atherosclerosis Disease"

_cells, 2021, doi:10.3390/cells10081985_

Round 1

Reviewer 1 Report

In the review manuscript titled ‘Contemporary lifestyle and Neutrophil Extracellular Traps: An emerging link in Atherosclerosis disease’, the authors have represented an elaborate and updated review of the role of neutrophils, particularly NETs in the pathophysiology of atherosclerosis. The review is an interesting read, the schematic diagrams convey the message adequately and no doubt will be of great interest to the readership of this journal and researchers exploring atherosclerosis and neutrophil functionality. I have a few suggestions to make, which in my opinion will add yet relevant information and improve the presentation of the manuscript.

  1. Instead of discussing the recent methodologies for exploring NETs in a text box, I suggest the authors present this information in a tabulated format giving the targets identified, methods employed, and briefly mentioning the pros and cons of each method.
  2. Although the authors have briefly mentioned the significance of neutrophil ROS in triggering NETosis, they have not discussed neutrophil NOS and NO derived free radicals in their review. I request the authors to provide a separate paragraph or section on neutrophil NOS in vascular inflammation, endothelial dysfunction and immune cell interaction which are relevant in the context of atherosclerosis.
  3. A discussion on the relative timing of neutrophil migration or extravasation at the site of vascular inflammation relative to monocytes and platelets will be useful.
  4. NETs-platelet interaction is a vital contributing factor promoting thrombin generation and atherothrombosis. I suggest the authors discuss this topic in a separate section and its relevance in atherosclerosis leading to thrombo-ischemic complications.
  5. The section ‘The interplay of neutrophil extracellular traps with established risk factors for cardiovascular disease’ is a bit lengthy and needs proper editing, which could be accomplished by deleting the general information on cardiovascular risk factors and atherosclerosis.

Reviewer 2 Report

In this review, the authors discuss the inflammatory process associated with the development of atherosclerotic lesions. The general focus is on neutrophils and the role of Neutrophils Extracellular Traps (NETs) as active players on atherosclerosis, from lesion initiation/progression to plaque rupture. They critically assess some hypotheses linking lifestyle-derived risk factors for cardiovascular diseases with increased NETosis and atherosclerosis progression.

Overall this a well-written review. 

Minor concern:

Box 1, NET visualization approaches: It would be appropriate to cite the paper by Olga Tatsiy & Patrick McDonald Front Immunol. 2018; 9: 2036. DOI: 10.3389/fimmu.2018.02036
